# REVISITING LOCALITY-SENSITIVE BINARY CODES FROM RANDOM FOURIER FEATURES

## ABSTRACT

The method of Random Fourier Feature (RFF) has been popular for large-scale learning, which generates non-linear random features of the data. It has also been used to construct binary codes via stochastic quantization for efficient information retrieval. In this paper, we revisit binary hashing from RFF, and propose Sign-RFF, a new and simple strategy to extract RFF-based binary codes. We show the locality-sensitivity of SignRFF, and propose a new measure called ranking efficiency to theoretically compare different Locality-Sensitive Hashing (LSH) methods, which provides a new systematic and unified framework for LSH comparison. It suggests that SignRFF should be preferred in the high similarity region. Experiments are conducted to show that SignRFF is consistently better than the previous RFF-based method, and also outperforms other data-dependent and deep learning based hashing methods with sufficient number of hash bits. Moreover, the proposed ranking efficiency aligns well with the empirical search performance.

## 1 INTRODUCTION

Developing efficient machine learning algorithms in large-scale problems has been an important research topic to deal with massive data. In this paper, we focus on efficient retrieval/search methods, specifically, by designing similarity-preserving binary representations of the data. That is, for each data vector $x \in \mathbb{R}^d$, we hash it into a length-$b$ binary 0/1 vector $h(x) \in \{0,1\}^b$, where the geometry of the data should be well preserved in the Hamming space. Searching with binary codes has been widely applied in many applications, such as large-scale image retrieval (Weiss et al., 2008; Gong & Lazebnik, 2011; Kulis & Darrell, 2009; He et al., 2013; Lin et al., 2015; Liu et al., 2016; 2017; Song et al., 2018; Yu et al., 2018; Yan et al., 2021). The benefits are two-fold. Firstly, it may largely reduce the memory cost for storing massive datasets, especially with high-dimensional data. Secondly, it can significantly speedup the retrieval process. For instance, the binary codes can be used to build hash tables (e.g., Indyk & Motwani (1998)) for sub-linear time approximate nearest neighbor (ANN) search. Moreover, in the Hamming space, we can also apply exhaustive search, which is much faster than computing the Euclidean distances, plus the technique of multi-index hashing (Norouzi et al., 2012) can further accelerate the exact Hamming search by orders of magnitude.

In general, binary hashing methods can be categorized into supervised and unsupervised approaches. In this work, we focus on the unsupervised setting. Locality-Sensitive Hashing (LSH) (Indyk & Motwani, 1998) is one of the early hashing methods leading to binary embedding. The LSH targeting at cosine similarity (Charikar, 2002), also known as SimHash, generates binary hash codes using random hyperplanes by taking the sign of *data-independent* random Gaussian projections. For the cosine similarity, LSH has strict theoretical guarantee on the approximation error and search efficiency, but typically requires relatively long codes to achieve good performance. Hence, many works have focused on learning *data-dependent* short binary hash codes, through different objective functions. Examples include Iterative Quantization (ITQ) (Gong & Lazebnik, 2011), Spectral Hashing (SpecH) (Weiss et al., 2008) and Binary Reconstruction Embedding (BRE) (Kulis & Darrell, 2009). Recently, some unsupervised deep learning based methods have been proposed in the computer vision community, many of which are, to some extent, "task-specific" for cross-modal/video/image retrieval, implemented based on some deep models like the autoencoder and VGG-16 (Liu et al., 2016; Do et al., 2017; Chen et al., 2018; Li et al., 2019; Yang et al., 2019; Hansen et al., 2020; Liu et al., 2020; Qiu et al., 2021). By taking the advantage of the complicated model structures (e.g., CNN layers), these deep methods show promising performance in many image retrieval tasks.

Compared with the data-dependent methods (including deep methods), LSH has three advantages:

- Although the data-dependent procedures can provide improved performance with fewer binary codes, a known undesirable effect of many of these mechanisms is the performance degradation when we increase the code length $b$, as reported in prior literature (Raginsky & Lazebnik, 2009; Joly & Buisson, 2011). In our experiments, we will show that this may also be an issue for deep learning based methods. On the other hand, the performance (i.e., search accuracy) of the data-independent LSH would keep boosting with larger $b$. In many scenarios, it is often the case that only using short codes (e.g., $\leq 128$ bits) cannot achieve a desirable level of search accuracy for practical purposes. In these cases, one may need to use longer codes anyway, where LSH could be more favorable.

- LSH is very simple to implement (only with random projections), while data-dependent methods require additional optimization/training which might be costly. Moreover, large deep learning models would typically need longer inference time, which might be infeasible in practice where the query speed is important.

- It is difficult to characterize the properties of the data-dependent methods theoretically, while LSH enjoys rigorous guarantees on the retrieval/approximation performance.

Therefore, LSH is still a popular hashing method with great research interest and many practical applications (Shrivastava & Li, 2014; Qi et al., 2017; Driemel & Silvestri, 2017; Chen et al., 2019; Zandieh et al., 2020; Lei et al., 2020; Daghaghi et al., 2021).

## 1.1 LOCALITY-SENSITIVE HASHING FROM RANDOM FOURIER FEATURES

Kernel methods have gained great success in many machine learning tasks (Schölkopf & Smola, 2002; Zhu & Hastie, 2001; Avron et al., 2017; Sun et al., 2018). However, standard kernel methods require the $n \times n$ kernel matrix, which becomes computationally expensive on large-scale datasets with massive data points. To this end, Rahimi & Recht (2007) proposes the method of random Fourier feature (RFF), which defines a feature map that approximates shift-invariant kernels by the linear inner products. As such, the RFF preserves the "non-linear locality structure" of the data. This leads to numerous applications in large-scale learning where one trains linear models on RFF to approximate training non-linear kernel machines (Yang et al., 2012; Chwialkowski et al., 2015; Sutherland & Schneider, 2015; Avron et al., 2017; Sun et al., 2018; Tompkins & Ramos, 2018).

Given the popularity of RFF, one natural step is to apply it to search problems. Raginsky & Lazebnik (2009) proposes to construct binary codes from the RFF using stochastic binary quantization. We call this method "SQ-RFF", whose convergence (as $b \to \infty$) and concentration can be theoretically characterized. The author showed that SQ-RFF can achieve competitive search accuracy with sufficient $b$, e.g., $b \geq 512$. Since RFF itself is a widely used tool for large-scale kernel learning, SQ-RFF could be a convenient/useful tool in practical scenarios where RFF has been generated for training kernel machines, and the data scientist wants to further use it for efficient near neighbor retrieval.

## 1.2 OUR CONTRIBUTIONS

Given the benefits of LSH and RFF, we revisit hashing methods for non-linear kernels, and improve the prior RFF based hashing method. Our approach is named as "SignRFF". Specifically,

- We theoretically compare several linear and non-linear LSH methods in terms of a novel measure called the *ranking efficiency*, which is defined based on the probability of retrieving a wrong/reversed similarity ranking. Under this unified metric, the proposed SignRFF is uniformly better than SQ-RFF. Moreover, the ranking efficiency also indicates that typically one should prefer SignRFF over linear LSH when the near neighbors are close to each other, which is the first systematic comparison of linear vs. non-linear LSH in literature.

- Empirically, we conduct near neighbor search experiments on benchmark image datasets to compare the proposed SignRFF with other popular hashing methods, which verifies the superiority of SignRFF over SQ-RFF. We validate that the ranking efficiency metric aligns well with the empirical results. Moreover, SignRFF outperforms various competing methods (including deep methods) with moderately large number of hash bits, indicating its advantage in applications to achieve a high search recall/precision.

## 2 PRELIMINARIES

### 2.1 LOCALITY-SENSITIVE HASHING (LSH)

In large-scale information retrieval, exhaustive search of the exact nearest neighbors is usually too expensive. A common relaxation in this setting is the Approximate Nearest Neighbor (ANN) search, where we return the "good" neighbors of a query with high probability. In this paper, we will consider the search problem with data points in $\mathbb{R}^d$. $\boldsymbol{X}$ denotes the database consisting of $n$ data points, and $\boldsymbol{q}$ is a query point. $\boldsymbol{x}, \boldsymbol{y}$ are two data points with $\rho = \cos(\boldsymbol{x}, \boldsymbol{y})$.

**Definition 2.1** ($\tilde{S}$-neighbor). *For a similarity measure $S : \mathbb{R}^d \times \mathbb{R}^d \mapsto \mathbb{R}$, the $\tilde{S}$-neighbor set of $\boldsymbol{q}$ is defined as $\{\boldsymbol{x} \in \boldsymbol{X} : S(\boldsymbol{x}, \boldsymbol{q}) > \tilde{S}\}$.*

**Definition 2.2** (($c, \tilde{S}$)-ANN). *Assume $\delta > 0$ is a parameter. An algorithm $\mathbb{A}$ is a $(c, \tilde{S})$-ANN method provided the following: with probability at least $1 - \delta$, for $0 < c < 1$, if there exists an $\tilde{S}$-neighbor of $\boldsymbol{q}$ in $\boldsymbol{X}$, $\mathbb{A}$ returns a $c\tilde{S}$-neighbor of $\boldsymbol{q}$.*

One popular method satisfying Definition 2.2 is the Locality-Sensitive Hashing (LSH).

**Definition 2.3** (Indyk & Motwani (1998)). *A family of hash functions $\mathcal{H}$ is called $(\tilde{S}, c\tilde{S}, p_1, p_2)$-locality-sensitive for similarity measure $S$ and $0 < c < 1$, if for $\forall \boldsymbol{x}, \boldsymbol{y} \in \mathbb{R}^d$ and hash function $h$ uniformly chosen from $\mathcal{H}$, it holds that: 1) If $S(\boldsymbol{x}, \boldsymbol{y}) \geq \tilde{S}$, then $P(h(\boldsymbol{x}) = h(\boldsymbol{y})) \geq p_1$; 2) If $S(\boldsymbol{x}, \boldsymbol{y}) \leq c\tilde{S}$, then $P(h(\boldsymbol{x}) = h(\boldsymbol{y})) \leq p_2$, with $p_2 < p_1$.*

A key intuition of LSH is that, similar data points with have higher chance of hash collision in the Hamming space. In this paper, we will specifically consider the LSH for the cosine similarity (Charikar, 2002). For a data vector $\boldsymbol{x} \in \mathbb{R}^d$, the LSH binary code is given by

$$\textbf{LSH:} \quad h_{LSH}(\boldsymbol{x}) = sign(\boldsymbol{w}^T \boldsymbol{x}), \tag{1}$$

where $\boldsymbol{w}^T$ is a standard Gaussian random vector. We use $b$ i.i.d. $\boldsymbol{w}_1, ..., \boldsymbol{w}_b$ to generate $b$ LSH binary codes. The collision probability between the codes of two data points $\boldsymbol{x}, \boldsymbol{y}$ is

$$P(h_{LSH}(\boldsymbol{x}) = h_{LSH}(\boldsymbol{y})) = 1 - \frac{\cos^{-1}(\rho)}{\pi}, \tag{2}$$

where $\rho = \cos(\boldsymbol{x}, \boldsymbol{y})$ is their cosine similarity. Note that $P(h_{LSH}(\boldsymbol{x}) = h_{LSH}(\boldsymbol{y}))$ is increasing in $\rho$, which, By Definition 2.3, is the key to ensure the locality sensitivity.

### 2.2 KERNELIZED LOCALITY-SENSITIVE HASHING (KLSH)

In this paper, we will consider the popular Gaussian kernel function defined for $\boldsymbol{x}, \boldsymbol{y} \in \mathbb{R}^d$ as

$$k(\boldsymbol{x}, \boldsymbol{y}) = \exp\left(\frac{-\gamma^2 \|\boldsymbol{x} - \boldsymbol{y}\|^2}{2}\right),$$

where $\gamma$ is a hyper-parameter. Let $\Psi : \mathbb{R}^d \mapsto \mathcal{F}$ be the feature map with the kernel induced feature space $\mathcal{F}$. To incorporate non-linearity in LSH, Kulis & Grauman (2009) proposes Kernelized Locality-Sensitive Hashing (KLSH), by applying LSH (1) in the kernel induced feature space $\mathcal{F}$, i.e., $h(\boldsymbol{x}) = sign(\boldsymbol{w}^T \Psi(\boldsymbol{x}))$. However, as in many cases (e.g., for the Gaussian kernel) the map $\Psi$ cannot be explicitly identified, constructing the random Gaussian projection vector $\boldsymbol{w}$ needs some careful design. KLSH approximates the random Gaussian distribution using a sufficient number of data points, by the Central Limit Theorem (CLT) in the Reproducing Kernel Hilbert Space. One first samples $m$ data points from $\boldsymbol{X}$ to form a kernel matrix $\boldsymbol{K}$, then uniformly picks $t$ points from $[1, ..., m]$ at random to approximate the Gaussian distribution. The hash code has the form

$$\textbf{KLSH:} \quad h_{KLSH}(\boldsymbol{x}) = sign(\sum_{i=1}^{m} \boldsymbol{w}(i) k(\boldsymbol{x}, \boldsymbol{x}_i)), \tag{3}$$

where $\boldsymbol{w} = \boldsymbol{K}^{-1/2} \boldsymbol{e}_t$, and $\boldsymbol{e}_t \in \{0, 1\}^m$ has ones in the entries with indices of the $t$ selected points. We see that the codes $h_1, ..., h_b$ are actually dependent, and the quality of using CLT to approximate Gaussian distribution is not guaranteed, especially in high-dimensional kernel feature space. Jiang et al. (2015) re-formulates KLSH as applying LSH on the kernel principle components

in the kernel induced feature space, resolving a theoretical concern of KLSH. Notably, since KLSH uses a pool of data samples to approximate the Gaussian distribution, the hash codes are in fact dependent in practical implementation. It is observed that the performance of KLSH also drops as $b$ increases (Joly & Buisson, 2011), similar to the behavior of many data-dependent methods. We provide more detailed explanation in Appendix C.

## 3 LOCALITY-SENSITIVE BINARY CODES FROM RANDOM FEATURES

Rahimi & Recht (2007) proposes random Fourier feature (RFF) to alleviate the computational bottle-neck of standard kernel methods. For a data vector $\boldsymbol{x}$, the RFF for the Gaussian kernel is defined as

$$\textbf{RFF:} \quad F(\boldsymbol{x}) = \cos(\boldsymbol{w}^T\boldsymbol{x} + \tau), \quad (4)$$

where $\boldsymbol{w} \sim N(0, \gamma^2 \boldsymbol{I}_d)$ and $\tau \sim Unif(0, 2\pi)$ where $\boldsymbol{I}_d$ is the identity matrix. It holds that

$$\mathbb{E}[F(\boldsymbol{x})F(\boldsymbol{y})] = k(\boldsymbol{x}, \boldsymbol{y})/2.$$

That is, the non-linear kernel similarity can be preserved in expectation by the linear inner product of RFF. To obtain good approximation, we generate $b$ independent RFFs for each data point using i.i.d. $\boldsymbol{w_1}, ..., \boldsymbol{w_b}$ and $\tau_1, ..., \tau_b$, which can be subsequently used for various learning tasks.

From now on, we will assume the data vectors are normalized to have unit $l_2$ norm: 1) this is a standard pre-processing step in practice; and 2) the theoretical guarantees of LSH (1) is only valid for the cosine similarity (i.e., normalized data). Nevertheless, our analysis can be easily extended to the general unnormalized case. With normalized data, the Gaussian kernel can be written as a function of $\rho = \cos(\boldsymbol{x}, \boldsymbol{y})$:

$$k(\rho) \equiv k(\boldsymbol{x}, \boldsymbol{y}) = e^{-\gamma^2(1-\rho)}.$$

To extend RFF to efficient search problems, Raginsky & Lazebnik (2009) designs a mapping $[0, 1] \mapsto \{0, 1\}$ for each RFF. For $\boldsymbol{x} \in \mathbb{R}^d$, the code is produced by stochastic quantization (SQ):

$$h_{SQ}(\boldsymbol{x}) = sign(F(\boldsymbol{x}) + \xi) = sign(\cos(\boldsymbol{w}^T\boldsymbol{x} + \tau) + \xi), \quad (5)$$

where $\xi$ is a random perturbation from $Unif(-1, 1)$, with two possible choices on the dependency. In Raginsky & Lazebnik (2009), the same $\xi$ is used for all data points. Another option in litera-ture is to use i.i.d. $\xi$ for each data point, such that the binary codes admit $\mathbb{E}[h(\boldsymbol{x})h(\boldsymbol{y})] = k(x, y)$. For example, it is used in Zhang et al. (2019); Li & Li (2021) for large-scale low-precision train-ing. We also tested this strategy in our experiments. The search performance is however very poor, because more variation is introduced. On the other hand, we actually do not need the exactly es-timation/recovery of the kernel values in search problems. Hence, in this paper we will consider the baseline using the same $\xi$ for all data samples as in Raginsky & Lazebnik (2009). The collision probability of the so-called "SQ-RFF" is given as follows.

**Theorem 3.1** (Raginsky & Lazebnik (2009)). *For the SQ-RFF (5), for $\forall \boldsymbol{x}, \boldsymbol{y} \in \mathbb{R}^d$, it holds that*

$$P_{SQ}(\rho) := P(h_{SQ}(\boldsymbol{x}) = h_{SQ}(\boldsymbol{y})) = 1 - \frac{8}{\pi^2} \sum_{s=1}^{\infty} \frac{1 - e^{-\gamma^2 s^2 (1-\rho)}}{4s^2 - 1}. \quad (6)$$

This method is locality-sensitive w.r.t. $\rho$ because of the fact that (6) is an increasing function of $\rho$. Hence, it is also locality-sensitive w.r.t. the kernel $k(\rho)$ by the monotonicity of $k(\rho)$.

**Proposition 3.2.** *The SQ-RFF (5) is $(\tilde{k}, c\tilde{k}, P_{SQ}(\rho_1), P_{SQ}(\rho_2))$-locality sensitive w.r.t. similarity measure $k(\cdot)$, where $\rho_1 = \log(\tilde{k})/\gamma^2 + 1$ and $\rho_2 = \log(c\tilde{k})/\gamma^2 + 1$.*

In the kernel approximation problem, stochastic rounding introduces higher variance due to the noise $\xi$, which hurts the kernel estimation accuracy (Li & Li, 2021). Later, we will see that a similar effect also presents in search problems. To this end, we propose our simple strategy—directly taking the sign of the RFF, i.e., applying deterministic quantization. Formally, the so-called "SignRFF" approach constructs the binary codes for $\boldsymbol{x}$ by

$$\textbf{SignRFF:} \quad h_{sign}(\boldsymbol{x}) = sign(F(x)) = sign(\cos(\boldsymbol{w}^T\boldsymbol{x} + \tau)). \quad (7)$$

Operationally, SignRFF is extremely convenient. At the first glance, it may appear a bit surprising that this simple scheme has not been studied in literature. We believe one of the reasons is that the theoretical correctness of SignRFF is hard to justify. Indeed, with some recent progress on RFF, we can show that SignRFF is locality-sensitive.

**Lemma 3.3** (Li & Li (2021)). *For two normalized data points $\boldsymbol{x}, \boldsymbol{y}$ with cosine $\rho$, let $F(\cdot)$ be the RFF defined as (4). The joint distribution of $z_x = F(x)$ and $z_y = F(y)$ is*

$$f(z_x, z_y | \rho) = \frac{\sum\limits_{k=-\infty}^{\infty} \left[ \phi_\sigma(a_x^* - a_y^* + 2k\pi) + \phi_\sigma(a_x^* + a_y^* + 2k\pi) \right]}{\pi \sqrt{1 - z_x^2} \sqrt{1 - z_y^2}}, \tag{8}$$

*where $a_x^* = \cos^{-1}(z_x), a_y^* = \cos^{-1}(z_y)$, and $\phi_\sigma(\cdot)$ is the p.d.f. of $N(0, \sigma^2)$ with $\sigma = \sqrt{2(1-\rho)}\gamma$. Furthermore, $\mathbb{E}[sign(F(\boldsymbol{x}))sign(F(\boldsymbol{y}))]$ is an increasing function of $\rho$.*

**Proposition 3.4.** *The SignRFF (7) is $(\tilde{k}, c\tilde{k}, P_{sign}(\rho_1), P_{sign}(\rho_2))$-locality sensitive w.r.t. to $k(\cdot)$, with $\rho_1 = \log(\tilde{k})/\gamma^2 + 1$ and $\rho_2 = \log(c\tilde{k})/\gamma^2 + 1$, with collision probability*

$$P_{sign}(\rho) := P(h_{sign}(\boldsymbol{x}) = h_{sign}(\boldsymbol{y})) = 2 \int_0^1 \int_0^1 f(z_x, z_y | \rho) dz_x dz_y, \tag{9}$$

*where $f(z_x, z_y | \rho)$ is the density function (8).*

*Proof.* By Definition 2.3 and the monotonicity of $k(\rho)$, it suffices to show that the collision probability, $P_{sign}(\rho)$, is increasing in $\rho$. This immediately follows from Lemma 3.3 that $\mathbb{E}[sign(F(\boldsymbol{x}))sign(F(\boldsymbol{y}))] = P_{sign}(\rho) - (1 - P_{sign}(\rho)) = 2P_{sign}(\rho) - 1$ is increasing in $\rho$. $\qquad\square$

## 4 THEORETICAL COMPARISON OF LSH METHODS: A NEW SCHEME

In our theoretical analysis, we assume for KLSH (Kulis & Grauman, 2009) that the Gaussian projection in the kernel induced feature space $\mathcal{F}$ is truly random. That is, KLSH "ideally" performs LSH in the kernel space: $h_{KLSH}(\boldsymbol{x}) = sign(\boldsymbol{w}^T \Psi(\boldsymbol{x}))$ where $\boldsymbol{w}$ is a random Gaussian vector with proper dimensionality. Since $\Psi(\boldsymbol{x})^T \Psi(\boldsymbol{y}) = k(\boldsymbol{x}, \boldsymbol{y})$, applying (2) in $\mathcal{F}$ we obtain

$$P_{KLSH}(\rho) := P(h_{KLSH}(\boldsymbol{x}) = h_{KLSH}(\boldsymbol{y})) = 1 - \frac{\cos^{-1}(e^{-\gamma^2(1-\rho)})}{\pi}. \tag{10}$$

**Proposition 4.1.** *KLSH is $(\tilde{k}, c\tilde{k}, 1 - \frac{\cos^{-1}(\tilde{k})}{\pi}, 1 - \frac{\cos^{-1}(c\tilde{k})}{\pi})$-locality sensitive w.r.t. $k(\cdot)$.*

Recall Definition 2.3 of the $(\tilde{k}, c\tilde{k}, p_1, p_2)$-LSH. It is known (Indyk & Motwani, 1998) that one can construct an LSH data structure with the worst case query time $\mathcal{O}(n^R)$, where $R := \frac{\log p_1}{\log p_2}$ is called the *LSH efficiency*, which has been used in literature to theoretically compare different LSH methods, e.g. SimHash vs. MinHash (Shrivastava & Li, 2014). However, we should note that the LSH efficiency is only based on a worst case analytical bound. Therefore, it may not well explain or predict the practical search performance (See Appendix A.1 for related results).

We propose a novel measure for evaluating the search accuracy of LSH schemes, namely, the *ranking efficiency*, that can better compare LSHs in practice. The definition is motivated by the observation that regardless of the concrete approach (e.g., building hash tables or exhaustive search), the nearest neighbor retrieval, to a large extent, essentially boils down to estimating the Euclidean (cosine) similarity rankings in the Hamming space. Suppose $\boldsymbol{x}$ and $\boldsymbol{y}$ are two points in the database, and $\boldsymbol{q}$ is a query with $\rho_x = \cos(\boldsymbol{q}, \boldsymbol{x})$, $\rho_y = \cos(\boldsymbol{q}, \boldsymbol{y})$. Assume $\boldsymbol{x}$ is the true nearest neighbor of $\boldsymbol{q}$, implying $\rho_x > \rho_y$. By the property of LSH, we know that the hash collision probability $p_x > p_y$. For an LSH hash function $h$, define the corresponding collision probability estimators as

$$\hat{p}_x = \frac{1}{b} \sum_{i=1}^b \mathbb{1}\{h_i(\boldsymbol{x}) = h_i(\boldsymbol{q})\}, \qquad \hat{p}_y = \frac{1}{b} \sum_{i=1}^b \mathbb{1}\{h_i(\boldsymbol{y}) = h_i(\boldsymbol{q})\}. \tag{11}$$

Now the problem becomes comparing $\hat{p}_x$ and $\hat{p}_y$ to estimate the true ranking of $p_x$ and $p_y$. We consider the event of obtaining a wrong similarity comparison from our estimation, i.e. $\hat{p}_x \leq \hat{p}_y$. Obviously, a higher probability implies worse search accuracy, as we are more likely to retrieve the wrong nearest neighbor $\boldsymbol{y}$. Denote

$$E_x = \mathbb{E}[\mathbb{1}\{h(\boldsymbol{x}) = h(\boldsymbol{q})\}], \quad E_y = \mathbb{E}[\mathbb{1}\{h(\boldsymbol{y}) = h(\boldsymbol{q})\}],$$

$$Cov(\mathbb{1}\{h(\boldsymbol{x}) = h(\boldsymbol{q})\}, \mathbb{1}\{h(\boldsymbol{y}) = h(\boldsymbol{q})\}) = \Sigma = \begin{pmatrix} V_x & V_{xy} \\ V_{xy} & V_y \end{pmatrix}.$$

By the CLT, we know that as $b \to \infty$, $\begin{pmatrix} \hat{p}_x \\ \hat{p}_y \end{pmatrix} \sim N(\begin{pmatrix} E_x \\ E_y \end{pmatrix}, \Sigma/b)$. This approximation would be good with a sufficiently large number of $b$, e.g., $b \geq 30$. In this regime, we can compute

$$P(\hat{p}_x \leq \hat{p}_y) = P(\hat{p}_x - \hat{p}_y \leq 0) = 1 - \Phi\left(\frac{\sqrt{b}(E_x - E_y)}{\sqrt{V_x + V_y - 2V_{xy}}}\right), \tag{12}$$

where $\Phi(\cdot)$ is the c.d.f. of standard normal distribution. For all the LSH methods studied in this paper, the $E_x$ and $E_y$ are simply the collision probabilities, and $V_x = E_x(1-E_x)$, $V_y = E_y(1-E_y)$ from the binomial distribution. Yet, the covariance $V_{xy}$ is difficult to be analytically computed. For simplicity, we assume that this term has same relative influence on the estimators of all approaches, thus dropping it in the formal definition of the *ranking efficiency*, which is given as below.

**Definition 4.1** $((\rho, c)$-Ranking Efficiency). *For a LSH method, let the hash collision probability at cosine $\rho$ and $c\rho$ be $E$ and $E_c$, respectively. Let $V = E(1-E)$, $V_c = E_c(1-E_c)$. The $(\rho, c)$-ranking efficiency is defined as $RE = \frac{E-E_c}{\sqrt{V+V_c}}$.*

**Remark 4.1.** *The $RE$ can also be defined in terms of the non-linear kernel $k(\cdot)$. We use cosine $\rho$ here to address both linear and non-linear LSH for generality. In most retrieval applications, we may concern more about high $c$ (e.g., $c = 0.95$) which corresponds to similar data points.*

In general, higher $RE$ implies smaller probability of the reversed estimated ranking in (12), which is more favorable. In Figure 1, we provide the comparison of ranking efficiency at different $\rho$, which covers the cases in our empirical study (Section 5). We observe some consistent patterns:

- **Non-linear LSH.** Firstly, compared with the other RFF-based approach SQ-RFF, the proposed SignRFF is uniformly more efficient at all $\rho$. In other words, in search tasks, the stochastic quantization of SQ-RFF introduces more "variation" that lowers the ranking efficiency. Compared with KLSH, SignRFF is more efficient when $\gamma$ is large (e.g., $\gamma > 2$) in all plots, while KLSH tends to be more efficient with small $\gamma$. In general, SignRFF is the best in terms of ranking efficiency.

- **When should we prefer non-linear LSH?** The ranking efficiency gives a novel theoretical guidance on the appropriate choice between linear and non-linear LSH methods. We see that, when the target $\rho$ is high (e.g., $\rho > 0.8$), with proper tuning, kernel methods (SignRFF and KLSH) can be more efficient than linear LSH. However, if the target $\rho$ is small (e.g., $\rho < 0.7$), linear LSH might be more favorable. In other words, SignRFF could be better than LSH on datasets where the near neighbors are close to each other with high similarity/short distance, which is usually the case for, e.g., image data.

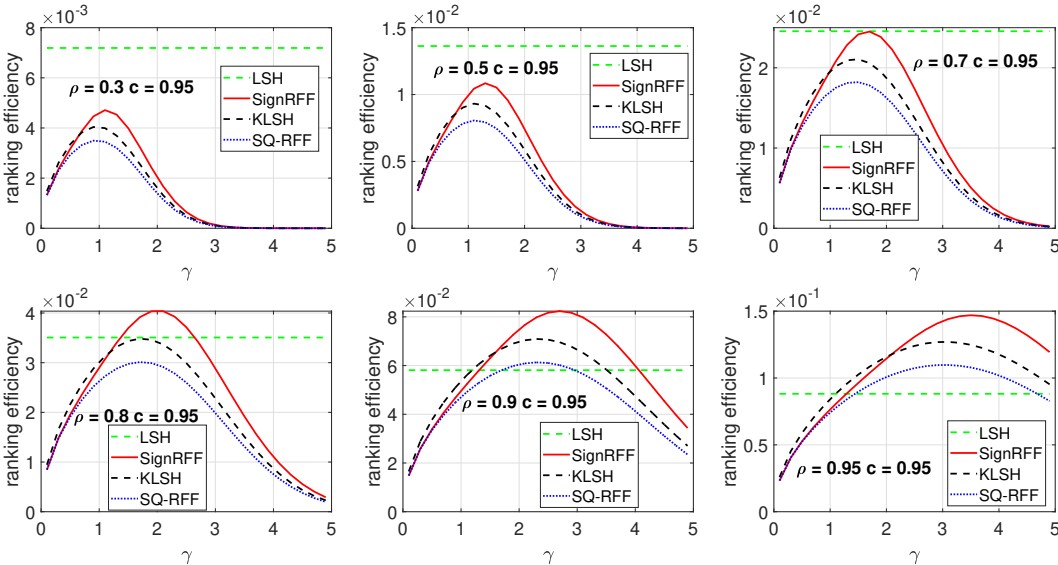

Figure 1: Ranking efficiency (Definition 4.1) of different LSH methods with various $\rho$, $c = 0.95$.

The ranking efficiency measures the search accuracy for given target $\rho$ value. In practice, a dataset contains many different pairs (with different $\rho$). Our experiments on real datasets in the next section show that the performance is largely consistent with the prediction based on the "average $\rho$" value.

## 5 EXPERIMENTS

In this section, we conduct image retrieval experiments on benchmark datasets to demonstrate the effectiveness of our approach and justify the theoretical comparison of ranking efficiency.

### 5.1 DATASETS, METHODS AND EVALUATION

**Datasets.** We use three popular benchmark datasets in image retrieval literature. The SIFT1M dataset (Jégou et al., 2011) contains 1M 128-dimensional SIFT image features, and 1000 query samples. The MNIST dataset (LeCun, 1998) contains 60000 hand-written digits. We randomly choose 1000 samples from the test set as the query points. The CIFAR dataset (Krizhevsky, 2009) contains 50000 natural images. We use the gray-scale images in our experiments, and randomly select 1000 test samples as the queries. In addition, when comparing our methods with VGG-16 (Simonyan & Zisserman, 2015) based deep methods, following prior literature (e.g.,Yang et al. (2019); Qiu et al. (2021)), we use the 4096-d features from the last fully connected layer of the pre-trained

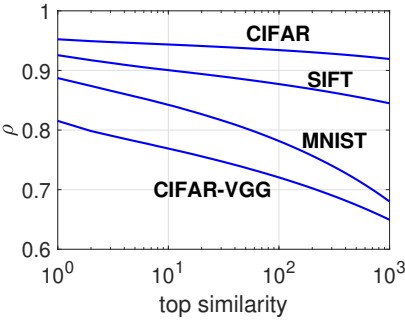

Figure 2: Average $\rho$ to $N$-th neighbor.

VGG-16 network as the input data for shallow methods for fairness. This dataset is called "CIFAR-VGG". On all datasets, the data points are normalized to have unit norm. In Figure 2, we report the average cosine similarity between queries to their $N$-nearest neighbor, from $N = 1$ to 1000.

**Methods.** We compare the following unsupervised hashing methods: 1) LSH (Charikar, 2002); 2) Iterative Qauntization (ITQ) (Gong & Lazebnik, 2011); 3) Spectral Hashing (SpecH) (Weiss et al., 2008); 4) Binary Reconstruction Embedding (BRE) with 1000 random samples for model training as suggested by Kulis & Darrell (2009); 5) KLSH with $m = 500$ random samples for formulating the kernel matrix and $t = 50$ samples for the CLT Gaussian approximation, which should be more accurate than $(300, 30)$ as recommended in Kulis & Grauman (2009); 6) SQ-RFF (Raginsky & Lazebnik, 2009); 7) Our proposed SignRFF. For kernel LSH methods 5) - 7), we tune the parameter $\gamma$ over $0.1 \sim 5$ and report the best $\gamma$ with highest average recall over multiple $b = 32 \sim 1024$. For each tested method, we generate binary codes and perform exhaustive search in the Hamming space.

**Evaluation.** For each query point, the ground truth nearest neighbors are set by ranking the top $N = 100$ smallest Euclidean distance (top-100 largest cosine similarity). After running each algorithm, the Hamming ranking returns $R$ estimated nearest neighbors to each query. We report the average recall over 1000 queries. Note that, recall@N is equivalent to precision@N in our setting. See Appendix A.2 for more detailed explanation and additional results on the retrieval precision.

### 5.2 RESULTS

In Figure 3, we report the recall@N (precision@N) against the number of binary codes $b$:

- On SIFT and MNIST, data-dependent methods (ITQ, SpecH, BRE) perform well with low bits, but the recall does not improve much after $b > 100 \sim 200$. Yet, this low bit region does not help much in practice, since the recall level is too low (e.g. $< 0.3$ on SIFT and CIFAR) to be satisfactory for real-world tasks. As we mentioned, this is a known limitation of these methods. When $b \geq 256$, LSH-type methods start to dominate, as expected. On CIFAR, SignRFF and KLSH outperform the data-dependent methods even with low bits.

- On all three datasets, SignRFF is substantially better than SQ-RFF with all $b$. Due to dependence, KLSH has higher recall than SignRFF when $b$ is small (e.g., $\leq 256$), but is beaten by SignRFF with more bits. The gap is significant and consistent when $b$ is as large as 512, where SignRFF achieves the highest recall on all three datasets.

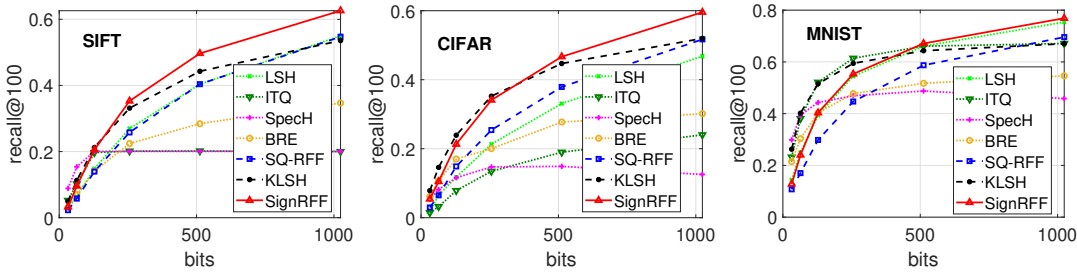

Figure 3: Recall vs. $b$. "recall@100" is the recall evaluated on the top-100 retrieved neighbors. Note that in our case, recall@100 is equivalent to precision@100.

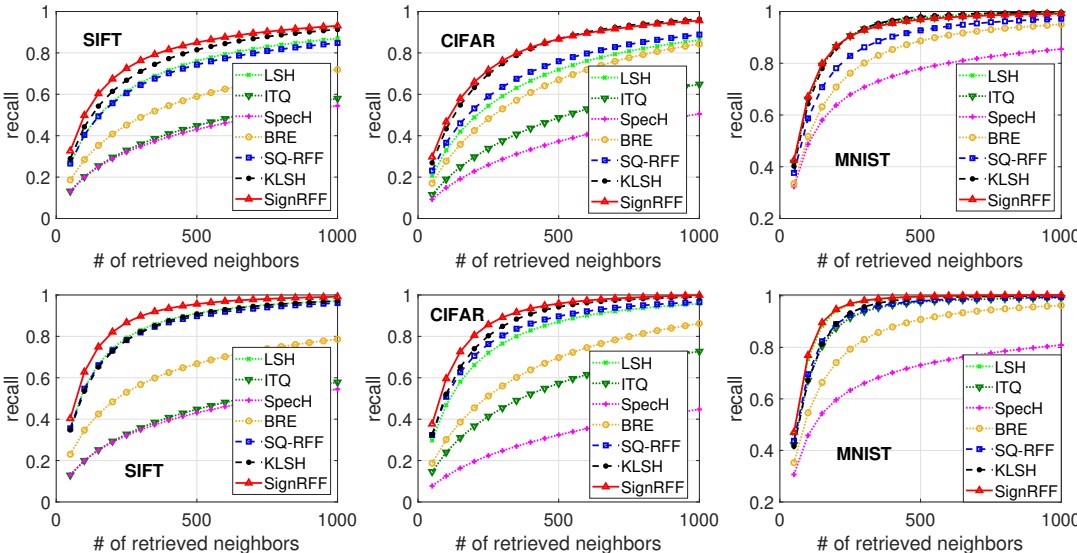

Figure 4: Recall vs. number of retrieved neighbors. **1st row:** $b = 512$. **2nd row:** $b = 1024$.

In Figure 4, we present the recall versus the number of retrieved neighbors, with $b = 512$ and $b = 1024$. SignRFF performs the best among all competing methods on SIFT and CIFAR. On MNIST, SignRFF performs similarly to LSH and better than the other methods with sufficient $b$.

**Ranking efficiency.** In Figure 5, we additionally provide the recall against the Gaussian parameter $\gamma$, to justify the effect of ranking efficiency discussed in Section 4. In Figure 1, the theoretical comparison suggests that KLSH would perform better with small $\gamma$, while SignRFF is more efficient larger $\gamma$, which matches the empirical evidence (e.g., 1 vs. 2.5 on SIFT). Moreover, as shown in Figure 2, the average $\rho$ between each query and its neighbors is around 0.8, 0.9 and 0.95 for MNIST, SIFT and CIFAR, respectively. Figure 1 predicts that compared with SQ-RFF (blue), LSH (green) would perform better on MNIST, similarly on SIFT, and worse on CIFAR, which aligns with

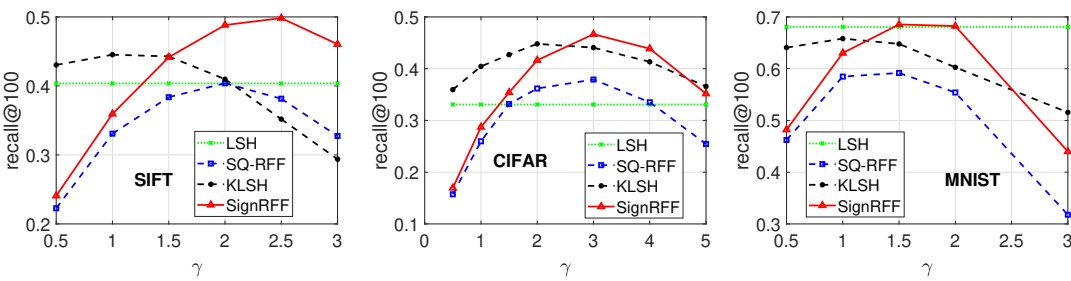

Figure 5: Recall@100 (precision@100) vs. $\gamma$, with $b = 512$.

Figure 3 and Figure 4. Therefore, although the behavior of KLSH slightly deviates from the theory due to the dependence among hash codes, our experiments in general verify the effectiveness of the ranking efficiency as an informative measure to compare different LSH methods.

### 5.3 COMPARISON WITH DEEP HASHING

We provide additional experiments on the CIFAR-VGG dataset with a recent unsupervised deep hashing method for image retrieval, the Contrastive Information Bottleneck (CIB) (Qiu et al., 2021), which uses VGG-16 pre-trained model as the backbone. We apply the same training setting as in Qiu et al. (2021): 60 epochs, 64 mini-batch size and Adam optimizer with 0.001 learning rate. We note that, CIB (as well as many other deep methods) is not designed to find the most similar data points. Instead, by using techniques like cropping and rotation in training CNN, these methods aim at robustly finding data points with same label as the query, which does not necessarily imply high similarity with the query when represented by the fc7 layer of VGG-16. Hence, to favor CIB, in this experiment we expand the range of true neighbors of a query to the top-1000 similar points in the database. See Appendix B for examples of the retrieved images and more discussion.

- From Figure 6, we see that CIB performs the best with 32 and 64 bits than LSH methods, which illustrates the benefit of deep hashing with short codes. However, the recall is only 0.3∼0.4, and does not improve with more bits, since simply increasing the size of last layer in the deep net typically would not elevate the model performance significantly. When $b \geq 256$, LSH, SignRFF and KLSH provide much higher recall than CIB.
- Among LSHs, KLSH performs the best on this task. SignRFF again consistently improves SQ-RFF, and matches the recall of KLSH with $b = 1024$. In addition, linear LSH slightly outperforms SignRFF, which is consistent with the theoretical comparison in Figure 1 as expected (the target $\rho$ is ∼0.7 for this task, where LSH is more efficient than SignRFF).

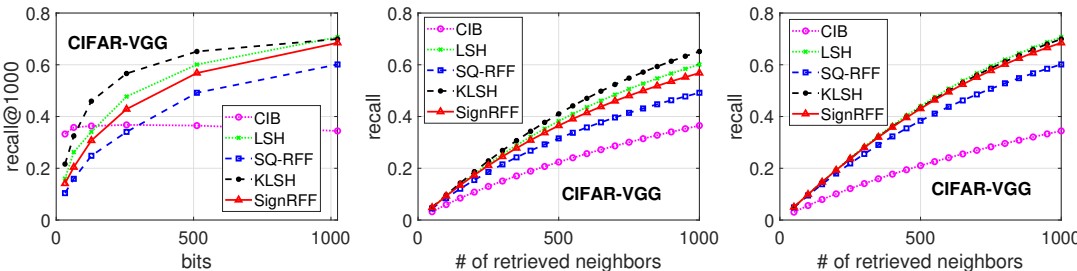

Figure 6: **Left panel:** Recall@1000 (precision@1000) vs. $b$. **Mid & Right panel:** Recall vs. # retrieved points, $b = 512, 1024$.

## 6 CONCLUSION

Random Fourier Feature (RFF) is a popular method in large-scale kernel learning, which has also been used for constructing binary codes for fast non-linear neighbor search (Raginsky & Lazebnik, 2009). In this work, we revisit RFF-based binary hashing, and propose a new strategy named Sign-RFF. We analyze its locality-sensitive property, and provide a new unified theoretical framework to compare different LSH methods, based on a measure called *ranking efficiency*. The metric implies that, we should prefer SignRFF over the linear LSH in the high similarity region when data points are close to each other. Empirically, we demonstrate that SignRFF is significantly better than the previous RFF-based approach, SQ-RFF, and outperforms data-dependent and deep hashing methods when the number of hash bits is moderately large. Moreover, the theoretical comparisons of the proposed ranking efficiency is consistent with the empirical results, suggesting that it could serve as a good theoretical tool for comparing the search performance of different LSH methods in practice.

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

## A    MORE ANALYTICAL FIGURES AND EXPERIMENTS

### A.1    COMPARISON OF LSH EFFICIENCY

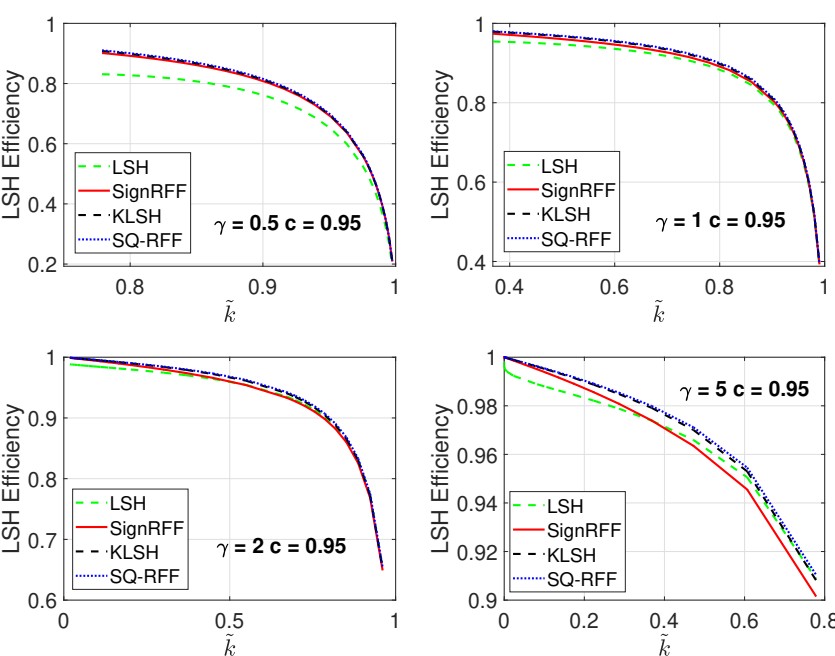

Figure 7: LSH efficiency of different LSH methods with various $\gamma$, $c = 0.95$. The $x$-axis $\tilde{k}$ is the Gaussian kernel value in alignment with Definition 2.3, Proposition 3.2, 3.2 and 4.1.

Recall Definition 2.3 of the $(\tilde{k}, c\tilde{k}, p_1, p_2)$-LSH. It is known (Indyk & Motwani, 1998) that one can construct an LSH data structure with the worst case query time $\mathcal{O}(n^R)$, where

$$R := \log p_1 / \log p_2$$

is called the *LSH efficiency*, which has been used in literature to theoretically compare different LSH methods, e.g. SimHash vs. MinHash (Shrivastava & Li, 2014). However, we found that in our case, the LSH efficiency does not provide much informative comparison of different LSH methods of interest. In Figure 7, we provide the LSH efficiency at multiple $\gamma$. Firstly, we see that the differences among the curves are very small. Basically, the figures tells us that SignRFF is always better than KLSH and SQ-RFF, but do not provide the comparison of SignRFF and KLSH regarding $\gamma$, as the ranking efficiency does. Also, the figures seem to suggest that SignRFF could (roughly) be better than LSH with large $\gamma$ and large $\rho$, but it does not give a good threshold at around $\rho = 0.7$ (validated by the experiments) as suggested by the ranking efficiency. Thus, LSH efficiency is insufficient to well predict the practical search performance.

### A.2    MORE EXPERIMENTS ON SEARCH PRECISION

Define $N$ as the number of ground truth neighbors. The search recall and precision are defined as

$$\text{recall@R} = \frac{\text{\# true neighbors in } R \text{ retrieved points}}{N},$$

$$\text{precision@R} = \frac{\text{\# true neighbors in } R \text{ retrieved points}}{R}.$$

Consequently, by definition, recall@N is equivalent to precision@N. Therefore, in our experiments, i.e., Figure 3, Figure 5 and Figure 6, the curves are also "precision@N". In Figure 8, we report more results on the search precision@R. As we can see, on SIFT, CIFAR and MNIST, the SignRFF method again performs the best as long as $b \geq 256$. The comparison is fairly consistent with the recall (precision) curves presented in Section 5.

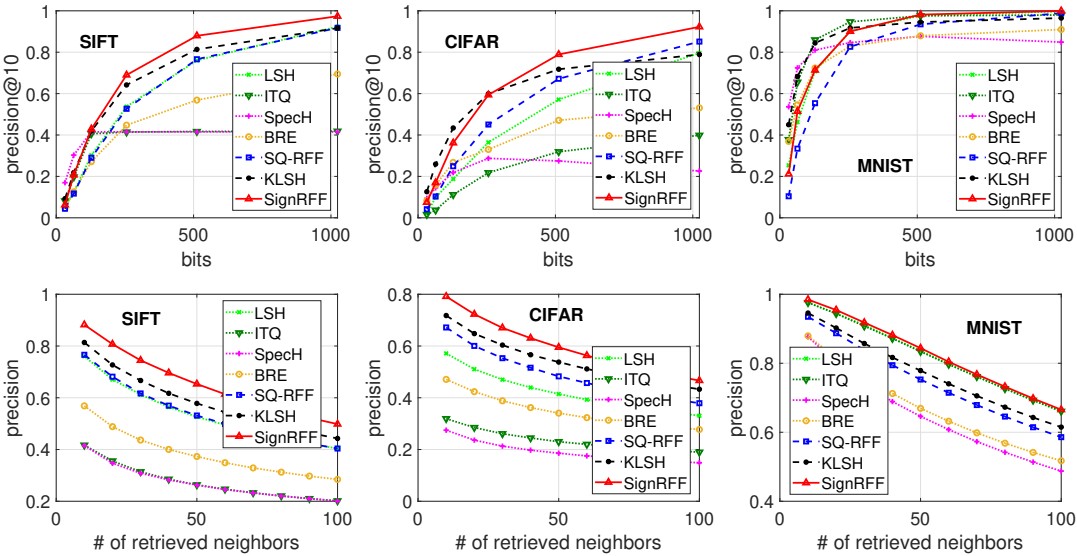

Figure 8: **1st row:** Precision@10 against $b$. **2nd row:** Precision vs. # of retrieved points, $b = 512$.

## A.3 DATA PROCESSING TIME

Another benefit of SignRFF is its simplicity in implementation. This can be reflected in the data processing time. Note that, the query time consists of two parts: the processing time (to generate query binary codes), and the search time (to compare the query codes to the database). In our experiments, since we adopt the standard Hamming ranking method, the search time to return a fixed number of retrieved points are the same for all algorithms. Hence, we plot the comparison of processing time (for 1000 queries), as shown in Figure 9. We observe that SignRFF and LSH are the most efficient methods. KLSH is notably slower than SignRFF. The two data-dependent methods, SpecH and BRE, are significantly slower than SignRFF. This reveals a potential advantage of the simplicity of SignRFF in practical retrieval systems.

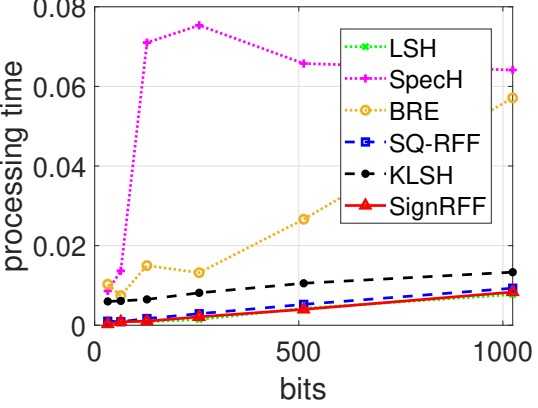

Figure 9: Data processing time (1000 queries).

## B    MORE IMAGE RETRIEVAL RESULTS ON CIFAR-VGG

In Section 5.3, we compare the locality-sensitive hashing methods with a deep learning based method, the Contrastive Information Bottleneck (CIB) (Qiu et al., 2021). CIB is built upon the pre-trained VGG-16 CNN model, whose objective is to generate binary representations such that images from the same class are close. Thus, same as many other deep hashing methods, the objective of CIB is slightly different from our setting (we find the most similar data points, without any label information), as being in the same class does not necessarily implies high similarity. In fact, in the empirical evaluation of many of these papers (e.g., Yang et al. (2019); Qiu et al. (2021)), the true neighbors are simply set as those data points with same label as the query. In our setting, we strictly follow the ranking to find the true neighbors with highest similarities. To better illustrate the difference, we present top-10 retrieval result of two queries (automobile and cat) in Figure 10. For SignRFF and KLSH, the retrieved images mostly have high similarity with the query, but may include some data points from other classes (e.g., truck vs. automobile, dog vs. cat). On the contrary, the retrieved images of CIB clearly have lower similarity with the query, but contain fewer other classes. This is largely because the VGG-16 is pre-trained by classification. Hence, in this experiment we set the true neighbors as the top-1000 similar data points, where CIB could be evaluated properly since most images from the same class would, at least, have not too low similarity.

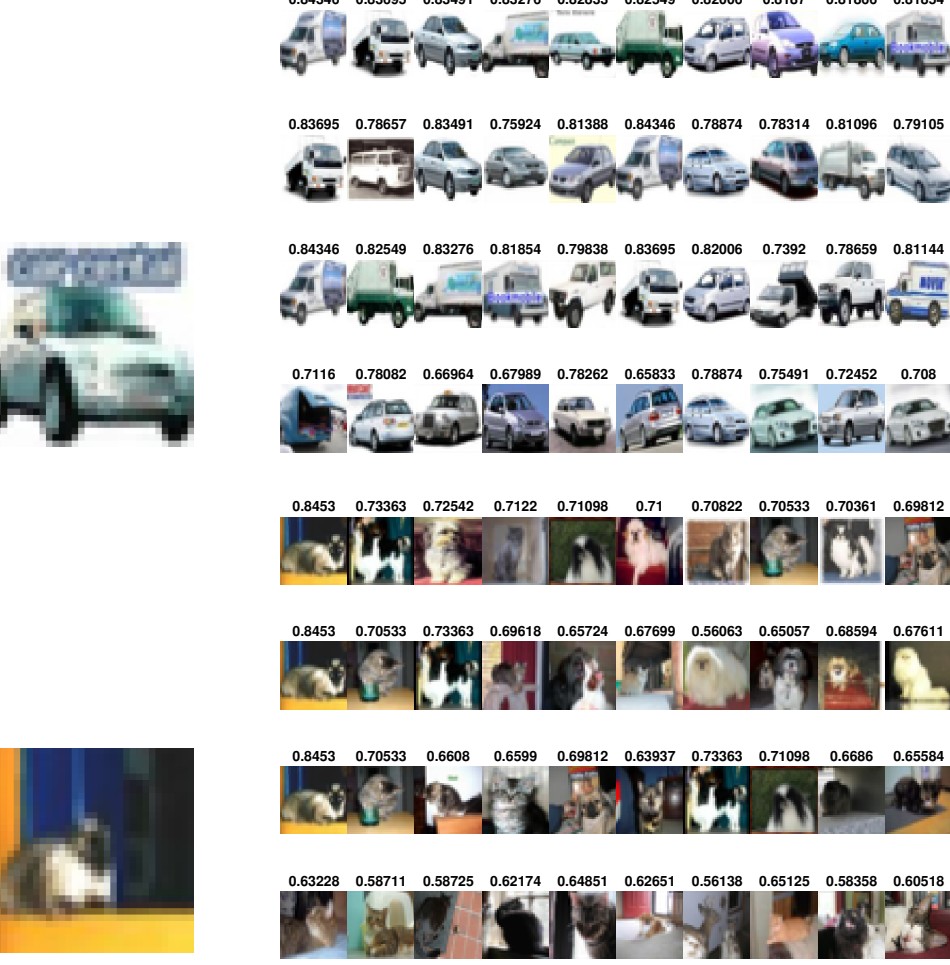

Figure 10: CIFAR Top-10 retrieved images (right) for two query points (left) with $b = 512$: Automobile and Cat. **1st row:** true nearest neighbors in terms of cosine similarity of the features extract from the last fc layer of VGG-16. **2nd row:** SignRFF. **3rd row:** KLSH. **4th row:** CIB. The number on each retrieved image is the cosine similarity to the VGG-feature of the query.

## C    IMPACT OF THE DEPENDENCE IN KLSH

In practical implementation, the hash codes of KLSH are dependent. We give an intuitive explanation on how this dependence affect the search performance.

Firstly, let us review the reason why data-independent methods with independent codes (LSH, SQ-RFF, SignRFF) can boost search accuracy with increasing $b$. Essentially, same as the intuition of the rank efficiency, comparing Hamming distance is equivalent to searching for the data points with highest estimated hash collision probability using $b$ codes. Let $\boldsymbol{x}, \boldsymbol{y}$ be two database points and $\boldsymbol{q}$ be the query, with $\rho_x > \rho_y$. This means the hash collision probability $p_x > p_y$. We estimate the probability by averaging the collision indicators:

$$\hat{p}_x = \frac{1}{b}\sum_{i=1}^{b}\mathbb{1}\{h_i(\boldsymbol{x}) = h_i(\boldsymbol{q})\}, \quad \hat{p}_y = \frac{1}{b}\sum_{i=1}^{b}\mathbb{1}\{h_i(\boldsymbol{y}) = h_i(\boldsymbol{q})\}. \tag{13}$$

For data-dependent methods, with sufficient $b$, (12) gives the probability of wrong raking (i.e., $\hat{p}_x < \hat{p}_y$). We see that it strictly decreases with larger $b$.

Recall the steps of KLSH. We first sample $m$ data points from database $\boldsymbol{X}$, denoted as $\tilde{\boldsymbol{X}}$, to form a kernel matrix $\boldsymbol{K}$. Then we uniformly pick $t$ points, denoted as $\tilde{\boldsymbol{X}}'$, from $[1, ..., m]$ at random to approximate the Gaussian distribution. After some algebra, the hash code has the form

$$\textbf{KLSH:} \quad h(\boldsymbol{x}) = sign(\sum_{i=1}^{m}\boldsymbol{w}(i)k(\boldsymbol{x}, \boldsymbol{x}_i)), \tag{14}$$

where $\boldsymbol{w} = \boldsymbol{K}^{-1/2}\boldsymbol{e}_t$, and $\boldsymbol{e}_t \in \{0, 1\}^m$ has ones in the entries with indices of the $t$ selected points. To generate multiple codes, we use the same pool of points $\tilde{\boldsymbol{X}}$, but with different choice of $e_t$, i.e., $\tilde{\boldsymbol{X}}'$ to approximate the Gaussian distribution.

**Boosted performance with small** $b$**.** For two hash codes $h_k$ and $h_{k-1}$ of $\boldsymbol{x}$, the $k(\boldsymbol{x}, \boldsymbol{x}_i)$ terms in (14) are the same. The $\boldsymbol{w}$'s are dependent since the points used for Gaussian approximation (i.e. $\tilde{\boldsymbol{X}}'$) may overlap. Thus, the conditional hash collision probability is usually larger than the unconditional one,

$$P(h_k(\boldsymbol{x}) = h_k(\boldsymbol{y})|h_{k-1}(\boldsymbol{x}) = h_{k-1}(\boldsymbol{y})) > P(h_k(\boldsymbol{x}) = h_k(\boldsymbol{y})),$$

and it may increase as $k$ get larger (intuitively, by dependence, more previous collisions implies higher chance of collision later on). Similarly,

$$P(h_k(\boldsymbol{x}) \neq h_k(\boldsymbol{y})|h_{k-1}(\boldsymbol{x}) \neq h_{k-1}(\boldsymbol{y})) > P(h_k(\boldsymbol{x}) \neq h_k(\boldsymbol{y})).$$

Thus, at the beginning, the more similar (dissimilar) the data pair is, the more the estimator $\hat{p}$ is upward (downward) biased. In other words, similar points would have further increased hash collision probability, while dissimilar points would have even lower chance of collision. This is the main reason that the empirical performance of KLSH is higher than theoretical prediction (where we assume independence) with short binary codes.

**Slow improvement with large** $b$**.** However, this is not the whole story. As more bits are used, there is less and less marginal information left. That is, the later generated codes would be more and more dependent on the previous ones. In an extreme case, at the point when all $\binom{m}{t}$ combinations of the $t$ points in $\tilde{\boldsymbol{X}}'$ have been used, the hash codes produced afterwards would all be the same as some previously generated ones, which would hardly improve the distance estimation anymore. This is why for KLSH, the recall curve becomes flat as $b$ increases.

