# OpenReview forum: "Revisiting Locality-Sensitive Binary Codes from Random Fourier Features"
_ICLR.cc/2022/Conference — ICLR 2022 Submitted_

### Official Review · Reviewer_tti8 · 2021-11-02

**Correctness:** 3
**Technical Novelty And Significance:** 2
**Empirical Novelty And Significance:** 3
**Recommendation:** 6
**Confidence:** 4

**Details Of Ethics Concerns:**

No.

**Main Review:**

Pros:
1. The authors develop a simpler hashing function for binary coding, which removes the random shifting $\xi$ compared with SQ-RFF. They further demonstrate the new hashing function is also locality-sensitive.

2. The new proposed ranking efficiency looks interesting. The authors provide some insights from the perspective of distortion or order-preserving (actual distance in original space vs. hamming distance after hashing) to illustrate the formula of ranking efficiency. They also use extensive results to justify the correctness of their formula.

3. The authors conduct comprehensive experiments to validate the effectiveness of SignRFF.

4. The paper is well organized and easy to follow.

Cons:
1. The critical concern of this paper is its contribution. The new hashing scheme SignRFF and the ranking efficiency make me feel like two separate works but combine together. Both of them lack further investigation and analysis. More details can be found from Points 2 & 3 below.

2. After introducing the new hashing function SignRFF and showing its locality-sensitive property, the authors did not further investigate its property. Based on Equation 9, we can compute the new $p_1$ and $p_2$ for $\rho_1$ and $\rho_2$. Can they be amplified compared with the $p_1$ and $p_2$ from SQ-RFF? They argue the stochastic rounding introduces extra variance caused by $\xi$, so can SignRFF reduce the variance? Can it reduce a lot or a little, or depend on other factors?

3. The authors claim the LSH efficiency ($R = \log p_1 / \log p_2$) is based on a worst-case analytical bound, but the LSH efficiency is still trendy. More investigations and comprehensive comparisons between the LSH efficiency and ranking efficiency can make it more convincing. Therefore, it is interesting to show the LSH efficiency of LSH, KLSH, SQ-RFF, and Sign-RFF with respective to $c$, $\rho$, and $\gamma$, and analyze whether the observed pattern from Figure 1 is consistent with that of the LSH efficiency.

4. In the experiments, since different methods use different LSH functions, their computation time may be quite different. As the query response time is important, can the authors also show the recall vs query time with respect to different b values?

Minor Comments or Typos:
1. In Figure 5, the recall vs. $\gamma$ for the data set Mnist is missing. Can the authors also show this curve in Figure 5?

2. In Page 5, the paragraph before Equation 11, "namely, the ranking accuracy, that can better compare LSHs in practice", "ranking accuracy" should be "ranking efficiency".

3. In Page 6, the $\hat{S}_y$ and $\hat{S}_x$ in Equation 12 should be $\hat{p}_y$ and $\hat{p}_x$, respectively.




**Summary Of The Paper:**

The paper revisits the binary hashing from Random Fourier Feature (RFF) for approximate nearest neighbor search. The authors propose a simple and effective RFF-based hashing method SignRFF and demonstrate its locality-sensitive property. They also introduce a new measure called ranking efficiency to evaluate the performance of different LSH methods theoretically. Extensive experiments on three real-life image data sets validate the effectiveness of SignRFF and the consistency of the ranking efficiency from theoretical evaluation.

**Summary Of The Review:**

In summary, this work looks interesting, but the concern about the contribution (details in Cons 1,2,3) justifies my current rating. I am glad to increase my rating if the authors can solve my questions. Thanks.

---

> ### Author Response · Authors · 2021-11-19
> **Response to Reviewer tti8**
>
> Dear Reviewer,
>
> We appreciate your valuable comments and suggestions. We have corrected the typos in the revision and added the result of MNIST in Figure 5.
>
> 1. **Contribution and motivation.** The roadmap of this paper is as follows. We first propose SignRFF, a simple yet unstudied hashing approach from RFF, and prove its correctness (locality sensitivity). Then, we compare the proposed SignRFF with other LSH methods, by the novel measure of ranking efficiency:
>
> (1) SignRFF is uniformly better than SQ-RFF;
>
> (2) KLSH performs be better with small $\gamma$, and SignRFF performs better with larger $\gamma$;
>
> (3) SignRFF tends to be better than linear LSH in the high ``target similarity'' region.
>
> Lastly, we show that empirical results well align with the theoretical comparison of ranking efficiency. Therefore, our second contribution (ranking efficiency) serves for justifying the advantage of our first contribution (SignRFF), and identifying scenarios where each method works better, from a systematic theoretical view. Thus, we believe that they are well related contributions.
>
>
> 2. Thanks for the nice question. In short: the ranking efficiency already incorporates both the mean and the variance information. As you kindly pointed out, given eq. (9) and eq. (6), we have already obtained the mean and variance of the collision probability estimator of SignRFF and SQ-RFF. We did not compare them separately because they, when investigated individually, are not informative enough for our purpose (ranking). Indeed, this is exactly why we proposed the ranking efficiency, which contains both first moment and second moment information. Note that the ranking efficiency is motivated by the approximation of reversed ranking probability. Thus, it serves well specifically for our goal to evaluate/compare the LSH search performance.
>
> Regarding the extra ''variance'', it is known in literature that the additional noise in SQ-RFF causes larger kernel estimation variance. In some sense, in the context of search (ranking) as in this paper, we could say that this extra noise leads to larger **''variation''**, by lowering the ranking efficiency. We have added more accurate statements on this point in the updated version, page 4 the paragraph before eq. (7) and page 6 paragraph ''Non-linear LSH''. We hope this answers your question.
>
> 3. Please kindly check Appendix A.1 in the revised paper for the LSH efficiency comparisons. First, the differences among the curves are very small. Basically, the figures tell us that SignRFF is always better than KLSH and SQ-RFF, but do not provide the comparison of SignRFF and KLSH regarding $\gamma$, as the ranking efficiency does. Also, the figures seem to suggest that SignRFF could (roughly) be better than LSH with large $\gamma$ and large $\rho$, but it does not give a good threshold at around $\rho=0.7$ (validated by the experiments) as suggested by the ranking efficiency. Thus, the proposed ranking efficiency is a better predictor of the practical search performance of LSHs.
>
> 4. Please see the updated Appendix A.3 for a time comparison. The query time consists of two parts: the processing time (to generate query binary codes), and the search time (to compare the query codes to the database). In our experiments, since we adopt the standard Hamming ranking method, the search time to return a fixed number of retrieved points are the same for all methods. Thus, we plot the comparison of processing time (for 1000 queries). As we can see, the SignRFF and LSH are the fastest methods. KLSH is notably slower than SignRFF. The two data-dependent methods, SpecH and BRE, are significantly slower. This reveals a potential advantage of the simplicity of SignRFF in practical retrieval systems.
>
> Thanks again for your feedback. We hope our response can adequately answer your questions.

---

> > ### Comment · Reviewer_tti8 · 2021-11-21
> > **Feedback to the authors**
> >
> > Thank you for your response and the new experiments.
> >
> > The authors have addressed many of my concerns, such as the LSH efficiency, the ranking efficiency, and the query time concern. The proposed ranking efficiency is now more convincing to me. The query time of SignRFF is also slightly lower than that of SQ-RFF.
> >
> > However, I am not very satisfied with the explanation of extra variance. Note that as the $\mathbf{w}$ and $\tau$ are randomized, both SignRFF and SQ-RFF need stochastic rounding (the sign quantization in Equations 5 & 7) even though the authors remove the random shift $\xi$. It will be more convincing if the authors can quantize the variance and use experiments to show how SignRFF can reduce it with/without $\xi$. This also makes me concerned about the novelty of SignRFF as it only simplifies the formula.
> >
> > In summary, as the authors address many of my concerns, I am happy to increase the rating to 6, but I think it is still a borderline paper. Thank you.

---

> > > ### Author Response · Authors · 2021-11-23
> > > **Clarification about the role of "Gap" probability and difference between SignRFF and SQ-RFF**
> > >
> > > Thanks for your feedback. We are glad that our rebuttal has addressed many of your questions. Here, we hope to use the opportunity to address your further questions:
> > >
> > > Firstly, we would like to explain more on the motivation and design of SignRFF.
> > >
> > > *''... This also makes me concerned about the novelty of SignRFF as it only simplifies the formula''*
> > >
> > > SignRFF and SQ-RFF are actually very different quantization approaches. Suppose we have the RFF of a data point $x$, denoted as $z=\cos(w^Tx+\tau)$ with $w\sim N(0,1)$, $\tau\sim Unif(0,2\pi)$. The definition of SQ-RFF is Eq. (5): $Q(z)=sign(z+\xi)$ with $\xi\sim Unif(-1,1)$. Here, we call it ''stochastic'' quantization (or stochastic rounding) because, the drifting term $\xi$ is equivalent to the following procedure: given $z$, assign $Q(z)=1$ with probability $|1+z|/2$ and $Q(z)=-1$ with probability $|1-z|/2$. Thus, it is a probabilistic quantizer. In contrast, the SignRFF directly takes the sign of the RFF $z$, so it is ''fixed''. Your question reminds us that this more straightforward description would help the readers better understand their differences. We will be happy to add this introductory explanation to the revision.
> > >
> > > We hope this clarification can also address your question about presenting the ''variance''. As one would expect, since the two quantizers (stochastic vs. fixed) are very different, the mean and variance of their collision probability estimators (Eq. (11)) are also different (as can be seen explicitly from Eq. (6) and Eq. (9) also).
> > >
> > > In the paper, the only one place that we mention the term ''variance'' is after Proposition 3.2:
> > >
> > > ''In the kernel approximation problem, stochastic rounding introduces higher variance due to the noise $\xi$, which hurts the kernel estimation accuracy (Li & Li, 2021).''
> > >
> > > This statement describes the finding in recent literature on the exact kernel approximation problem, which is different from the goal of our paper, i.e.,  approximate near neighbor (ANN) search， which  cares about the relative ranking/ordering. For ANN problem, what matters is the "Gap" between probabilities, which we will further elaborate as follows:
> > >
> > > For data points $x$ and $y$, the collision probability estimator $\hat p$ is given in the form of Eq. (11). For ANN search, by LSH property, more similar data pair would have higher hash collision probability. Therefore, the problem turns into:
> > >
> > > given two pairs of data points with true collision probability $p_1>p_2$ (associated with their true similarity), we want to find which one is larger using the estimates $\hat p_1$ and $\hat p_2$.
> > >
> > >
> > > Therefore, intuitively, we want the ''Gap'' between $\hat p_1$ and $\hat p_2$ to be large, such that there is small chance for us to get $\hat p_1<\hat p_2$ wrongly. In fact, if we examine the classical *LSH efficiency* $R=\frac{\log p_1}{\log p_2}$ (the smaller the better) mentioned in Section 4, it also measures this ''gap'' in some way. For example, for a given $p_1$, smaller $p_2$ (larger gap) would give smaller $R$. Our proposed ranking efficiency directly approximates the reversed ranking probability, to better characterize this gap. From Definition 4.1 we see: 1) if two estimators have same mean, then the one with smaller variance would have higher ranking efficiency; 2) if two estimators have same variance, then the one with larger ''Gap'' ($|p_1-p_2|$) would have higher efficiency. Thus, the definition naturally combines both the mean and variance. Our results reveal that by eliminating the drifting term $\xi$, SignRFF always has higher ranking efficiency than SQ-RFF.
> > >
> > > We hope our reply explains the essential difference between the stochastic SQ-RFF and the fixed SignRFF quantization. We also hope that our clarification on the variance vs. ranking efficiency can well answer your concern and provide better intuition of the proposed ranking efficiency.
> > >
> > > We sincerely thank you again for the detailed comments and the nice suggestions.

---

### Official Review · Reviewer_TrWv · 2021-11-02

**Correctness:** 3
**Technical Novelty And Significance:** 2
**Empirical Novelty And Significance:** 1
**Recommendation:** 5
**Confidence:** 3

**Main Review:**

Strengths:
1. The proposed SignRFF is simple. It is a simplified version of SQ-RFF by taking out the random perturbation.
2. A new measure called the ranking efficiency is proposed to measure the practical performance rather than the worst case performance of LSH methods. This measure makes sense to consider the probability of retrieving a wrong/reversed similarity ranking.
3. The paper is well written and easy to follow.

Weaknesses:
1. The technical contribution is the comparison of different LSH methods with the new defined ranking efficiency measurement. While the ranking efficiency measure is computed with theoretical analysis and some approximations, the comparison is conducted empirically. Thus, I am concerned about the author's claim of this contribution to be theoretically comparison between LSH methods.
2. While the experimental results in Sec 5.2 look good, the results in Sec 5.3 (Figure 6) suggest the proposed method falls behind the KLSH. Any explanation for such performance gap or solution to remedy the problem would make the experiments and conclusion convincing.
3. The authors argue that using short codes (less than 128 bits) cannot achieve a desirable level of search accuracy for practical purposes. The authors show the variation of recall w.r.t different bits in Figure 3, but fail to show whether the studied bit length is the desirable one in practical situations. Also, the precision is expected to be reported in this study.

Minor:
The \hat{S} is not defined in (12)
There are research papers that study deep learning based hashing methods before 2019, such as
Deep Hashing via Discrepancy Minimization, 2018
Simultaneous feature aggregating and hashing for large-scale image search, 2017
Deep supervised hashing for fast image retrieval, 2016
Simultaneous feature learning and hash coding with deep neural networks, 2015
Supervised hashing for image retrieval via image representation learning, 2014
it's better to mention these works when introducing hashing with deep models.




**Summary Of The Paper:**

In this paper, the authors propose a simple strategy to extract random Fourier feature based binary codes. The authors also propose a new measure, ranking efficiency, to compare the search accuracy over two datapoints for locality-sensitive hashing methods. Experiments are conducted on several image retrieval datasets to validate the effectiveness of the proposed SignRFF.

**Summary Of The Review:**

My research experience is on design hashing for computer vision tasks and I am familiar with work on LSH.
The paper is overall well organized. The technical contribution is the derivation of the ranking efficiency measurement, which is lower than the bar of ICLR. There are some flaws as identified in the weaknesses.

---

> ### Author Response · Authors · 2021-11-19
> **Response to Reviewer TrWv**
>
> Dear Reviewer,
>
> Thanks for your valuable feedback and detailed comments. We have corrected the typo. Thanks also for the related references on deep hashing. We have cited many of them in the revision.
>
> 1. Figure 1 is the theoretical ranking efficiency of different LSH methods, all calculated exactly based on the formula in Definition 4.1. The ''theoretical comparison'' in the paper refers to this Figure 1. We drew some comparisons based on Figure 1 theoretically:
>
> (1) SignRFF is uniformly better than SQ-RFF;
>
> (2) KLSH performs better with small $\gamma$, and SignRFF performs better with larger $\gamma$;
>
> (3) SignRFF tends to be better than linear LSH in the high ``target similarity'' region,
>
> and validated them by experiments in section 5 empirically. Thus, the main comparison is given by theory (Figure 1), and the experiments are just to justify the theory.
>
> 2. As we mentioned in Section 2.2, before our results, it is well-known in literature that KLSH would particular perform better with short codes, and worse with longer codes, deviating from the expected behavior of LSH. We provided some explanation in Section B in the original submission (Section C in the updated version). Overall, this is because in practice, the hashes of KLSH are dependent. And also because of this, theoretically analyzing the practical implementation of KLSH appears difficult. Nevertheless, in Figure 6, we see that SignRFF tends to outperform KLSH if $b>1024$. On SIFT, CIFAR and MNIST, SignRFF is substantially better than all other competing methods when $b\geq 256$.
>
> 3. **Precision.** Thanks for the suggestion. In our experiments (Figure 3), we set $N=100$ ground truth neighbors. Therefore, the reported recall@N is in fact equivalent to precision@N. Recall is defined as $\frac{|A \cap B|}{|B|}$ and precision is $\frac{|A \cap B|}{|A|}$, where $A$ is the retrieved set and $B$ is the ground truth set. In our case, Both $A$ and $B$ equal $N$, so recall$@N=$ precision$@N$ here. Same argument holds for Figure 6 where $N=1000$. Yet, as you kindly suggested, in the updated version, we have presented more results on the search precision in Appendix A.2.
>
> In practice, the ''sufficient'' search recall/precision level depends on different applications. For example, practically speaking, we found some reports that examined the recall and precision of some commercial search engines:
>
> Precision and Relative Recall of Search Engines: A Comparative Study of Google and Yahoo, B.T. Sampath Kumar, J.N. Prakash.
>
> Evaluating Search Effectiveness of Some Selected Search Engines, M. S. Bute et al.
>
> The recall and precision of many search engines should usually reach at least 60\%$\sim$70\%. While this is of course not an universal law, it is certainly true that higher recall/precision is almost always more favorable in practice. In this case, using SignRFF with relatively long codes could be a good choice.
>
> Thanks again for your valuable feedback. We hope our rebuttal can well address your questions.

---

> > ### Comment · Reviewer_TrWv · 2021-11-24
> > **Response to Authors**
> >
> > 1. It is necessary to point out the 'theoretical comparison' is based on the observation from Figure 1. The curves in Figure 1 only cover some specific cases. And it is not guaranteed the conclusion can be generalized to other cases.
> >
> > 2. Any explanation about the comparison between KLSH and the proposed method?
> >
> > 3. There is only one case that recall@N equals to precision@N. That is when all the retrieved results are correct. Obviously, this is not the case in Figure 3. If the reported curves in Figures is calculated based on the top N returned results, it should be precision rather than recall.
> >
> >     It is still not clear whether the evaluated bit length is practical or not.
> >
> > Given the existing response, I would like to keep my rating at 5. And I am open for further discussions. Thanks.

---

> > > ### Author Response · Authors · 2021-11-30
> > > **Response to ''Response to Authors ''**
> > >
> > > Dear Reviewer,
> > >
> > > Thanks for your feedback.
> > >
> > > 1. The $\rho$ values presented in Figure 1 correspond to the cases in our experiments, i.e., Figure 2. In the anonymous link,
> > >
> > > https://anonymous.4open.science/r/ICLR_SignRFF_extra_plots-C04A/extra_plots.pdf
> > >
> > > we provide additional theoretical comparisons of ranking efficiency at more $\rho$ values ($\rho=0.01\sim 0.95$) and $c=0.5,0.7,0.95$ (in Figure 1, Figure 2, Figure 3, respectively). For the convenience of comparison, we plot the ratios, SignRFF/KLSH and SignRFF/SQ-RFF (i.e., values > 1 mean improvements by using our proposed SignRFF). We can see that the three  Figures (for three $c$ values) are actually very similar.
> > >
> > > In each figure, for a fixed $\rho$, we can see that the ratios have similar trends in that they increase with increasing $\gamma$.
> > >
> > > We hope these three figures could help clarify your concern of "The curves in Figure 1 only cover some specific cases. And it is not guaranteed the conclusion can be generalized to other cases."  When preparing for the paper, we had observed similar trends when we plotted more figures, but we agree with you that it should always help to provide more plots (some of which could be in placed in the Appendix). Thank you for the kind suggestion. Please let us know if you hope to see even more plots.
> > >
> > > 2. Conceptually, KLSH applies LSH in the kernel induced feature space, while our SignRFF is based on the explicit feature map of random Fourier features (RFF) by approximating the Fourier integral associated with the kernel function. Theoretically, the comparison of the search performance can be seen from Figure 1 and empirically verified by Figure 5. From our experiments, in terms of practical search accuracy, KLSH performs better with short codes due to the dependency (as we explained in the paper). SignRFF outperforms KLSH with larger $b$.
> > >
> > >
> > > 3.  We did quite get what you meant by pointing that some issues regarding precision/recall. Since they are fundamental measures, we hope to resort to  https://en.wikipedia.org/wiki/Precision_and_recall for the definitions of recall and precision:
> > >
> > > $\text{recall@R}=\frac{\text{number of true neighbors in top R retrieved points}}{N},$
> > >
> > > $\text{precision@R}=\frac{\text{number of true neighbors in top R retrieved points}}{R}.$
> > >
> > > You commented that
> > >
> > > "There is only one case that recall@N equals to precision@N. That is when all the retrieved results are correct."
> > >
> > > We do not quite understand this comment.  "When all the retrieved results are correct", then precision is 100\%, but the recall will vary depending on how many items will be returned. Of course, since in our experiment we also return exactly $R=N$ items, they are both 100\% in this case.
> > >
> > >
> > > You also commented that
> > >
> > > "If the reported curves in Figures is calculated based on the top N returned results, it should be precision rather than recall."
> > >
> > > Again, in the experiments, since we return $R=N$ items (to simplify the plots), precision = recall. If you prefer that we call it "precision" in Figure 3, we are happy to comply too.
> > >
> > >
> > > We should also emphasize that we did not invented the evaluation metric. It is used in, e.g.,
> > >
> > > [1] - Iterative quantization:  A procrustean approach to learning binary codes, Gong et al., CVPR 2011.
> > >
> > > [2] - K-means Hashing: an Affinity-Preserving Quantization Method for Learning Binary Compact Codes, He et al., CVPR 2013.
> > >
> > > Thank you very much. We hope we have addressed your additional concerns. Please let us know if we could provide further clarifications.

---

### Official Review · Reviewer_Mduj · 2021-11-03

**Correctness:** 4
**Technical Novelty And Significance:** 3
**Empirical Novelty And Significance:** 3
**Recommendation:** 6
**Confidence:** 2

**Main Review:**

1. The proposed ranking efficiency measure gives an objective measure of different Locality-Sensitive Hashing method.
2. The proposed hashing method was able to surpass other method. It is impressive given the simple modification based on SQ-RFF;



**Summary Of The Paper:**

This paper proposed an improved version of Random Fourier Feature (RFF) based binary feature coding method as well as a new measure(ranking efficiency) to compare different Locality-Sensitive Hashing (LSH) methods. Under the new measure, the proposed method is better than previous methods. The authors also validated by visualization the consistency between theoretical comparisons based the proposed measure and the empirical results.

**Summary Of The Review:**

The author does a great job explaining the idea, concepts, procedures and experiments.

---

> ### Author Response · Authors · 2021-11-19
> **Response to Reviewer Mduj**
>
> Dear Reviewer,
>
> We appreciate your nice summary of our work and the support! We have revised the submission to include more contents and experiments that may interest you. Thank again for your nice comments.

---

### Official Review · Reviewer_3RfW · 2021-11-03

**Correctness:** 3
**Technical Novelty And Significance:** 2
**Empirical Novelty And Significance:** 2
**Recommendation:** 3
**Confidence:** 3

**Main Review:**

I have the following concerns of the paper:

* limited method novelty: the proposed method SignRFF (eq. (7)) simply replaces the stochastic rounding into a deterministic form. In the whole section 3, only Proposition 3.4 is new, which proves that SignRFF is also locality-sensitive.

* The empirical observation "the results also reveals that LSH-type methods are better than data-dependent and deep hashing methods when the code length is moderately large" is not new as well, [1] has the exactly same, and comprehensive study.

* The empirical performance of SignRFF is not very competitive against other LSH methods.

* The adopted hamming ranking retrieval method is not very efficient. It'd be better to use hashing table(s) for evaluating retrieval performance.


[1] A Revisit of Hashing Algorithms for Approximate Nearest Neighbor Search, TKDE





**Summary Of The Paper:**

The paper improves RFF-based hashing methods, and verfies it empirically.

**Summary Of The Review:**

Given the limited method novelty and empirical findings, I'd recommend a reject.

---

> ### Author Response · Authors · 2021-11-19
> **Response to Reviewer 3RfW**
>
> Dear Reviewer,
>
> Thanks for your valuable feedback.
>
> 1. **Novelty.** The main motivations of our work are:
>
> (i) Revisit the RFF-based LSH and propose the simpler yet much better strategy SignRFF, which has not been formally investigated before.
>
> (ii) Propose the measure of ranking efficiency to provide a unified metric to better compare different LSH methods in practice. The empirical search performances align very well with the theoretical comparison of this measure.
>
> In particular, we would like to highlight the novelty and significance of the second contribution. As far as we know, there has been no systematic comparison of linear LSH with non-linear LSH in literature, especially from a theoretical viewpoint. Our proposed ranking efficiency matches very well with the empirical results, and it importantly identifies the scenarios where non-linear LSH could work better and cases where it could not. We believe this could be an interesting contribution to the field of LSH.
>
> 2. Thanks for pointing this out. We did not intend to claim ''LSH-type methods are better than data-dependent and deep hashing methods when the code length is moderately large'' as our new contribution, as we in fact have mentioned it in the introduction before Section 1.1. We have adjusted the statement in Section 1.2, highlighting the advantage of SignRFF over other methods.
>
> 3. **Performance of SignRFF.** Firstly, on SIFT, CIFAR and MNIST, SignRFF has a clear advantage over all other methods when $b$ is sufficient. Especially, it significantly improves SQ-RFF, the prior method based on RFF, which is the most related baseline of our method. Secondly, on CIFAR-VGG with smaller ''target similarity'' (Figure 2), SignRFF becomes a little bit worse than LSH. However, as we mentioned in the answer to question 1, *our primary goal is not to prove that SignRFF is the ''best'' method in all cases*. In fact, as the theoretical comparison of ranking efficiency suggests, when the target similarity is small (e.g., around 0.7 for CIFAR-VGG), linear LSH would be better than SignRFF. Hence, the empirical performance actually (again) well validates the effectiveness of the proposed ranking efficiency as a theoretical indicator for search performance.
>
> 4. **Hamming search vs. hash table.** Thanks for the suggestion. We use the simpler Hamming search to better align with the setting and derivation of the ranking efficiency in Section 4, which is then convenient for us to justify the theory by experiments. Using hash tables requires more complicated operations (and more tuning), so the empirical validation would not be very straightforward.

---

### Author Response · Authors · 2021-11-19
**General response and paper revision**

Dear Reviewers and Area Chairs,

We sincerely appreciate your effort in reviewing our paper, and the valuable comments and suggestions that helped us improve the quality of the paper. We have revised the submission (please kindly check the updated pdf) for some better clarification, and additional analytical figures and experiments. We also updated the abstract and introduction to better highlight our main contributions and novel results:

(i) we revisit binary hashing from RFF, propose the simple SignRFF method, and show its locality sensitivity; empirically, it outperforms most competing methods with moderate number of bits.

(ii) we propose the ranking efficiency, which can be used as a theoretical measure for a novel unified framework to compare different LSH methods in terms of search performance. The theoretical comparison implies several practical implications (e.g., we should prefer SignRFF over linear LSH in high similarity region) that are confirmed by our experiments.

We hope that our rebuttal and revision can adequately address your questions and concerns. Thanks again for your valuable feedback.

---

### Decision · Program_Chairs · 2022-01-20

**Decision:**

Reject

**Comment:**

Thanks for your submission to ICLR.

This paper considers binary hashing schemes, and makes two related contributions.  First, it analyzes a simple extension to SQ-RFF; second, it introduces and analyzes a novel metric for ranking called ranking efficiency.  Some experiments are also performed on standard data sets.

This is very much a borderline paper, and could go either way.  I took a close look at the paper to offer my opinions in addition to the reviewers.  The paper itself is well written and seems to be correct.  I do like the simplicity of the proposed SignRFF method as well as the ranking efficiency measure.  However, the contributions are somewhat limited, and it's in an area that hasn't seen much work in the last several years (this paper mainly builds off of methods from 10+ years ago).  Further, it doesn't seem that methods such as KLSH and SignRFF are used much in practice, so I don't know if this will have substantial impact.  So while it's a reasonably interesting paper with some nice insights, I think it falls just below the acceptance threshold for me.